# Influence of Age and Sex on Post-Craniotomy Headache

**DOI:** 10.3390/biomedicines12081745

**Published:** 2024-08-02

**Authors:** Jong-Ho Kim, Sung-Woo Han, Young-Suk Kwon, Jae-June Lee, Jong-Hee Sohn

**Affiliations:** 1Department of Anesthesiology and Pain Medicine, Chuncheon Sacred Heart Hospital, Hallym University College of Medicine, Chuncheon 24252, Republic of Korea; poik99@hallym.or.kr (J.-H.K.); gettys@hallym.or.kr (Y.-S.K.); 2Institute of New Frontier Research Team, Hallym University College of Medicine, Chuncheon 24252, Republic of Korea; hsw4070@naver.com; 3Department of Neurology, Chuncheon Sacred Heart Hospital, Hallym University College of Medicine, Chuncheon 24252, Republic of Korea

**Keywords:** post-craniotomy headache, age, sex, headache, craniotomy

## Abstract

Post-craniotomy headache (PCH) is a common postoperative complication, and some of these patients progress to chronic PCH (CPCH). We aimed to identify clinical variables associated with PCH and its progression to CPCH, especially possible associations between age and sex differences. Therefore, we examined clinical information on PCH using the Clinical Data Warehouse over 10 years. Of the 1326 patients included, 927 patients (69.9%) experienced PCH. In multivariate analysis for PCH, age was inversely related to risk (*p* = 0.003), and being female showed a significant association with an increased risk of PCH (*p* = 0.002). There was also a significant inverse relationship between age and severity of the worst headache, with younger female patients reporting greater severity of the worst headache (*p* < 0.001). Of the 927 patients who experienced PCH, 319 (34.4%) progressed to CPCH. Sex was a significant factor, with females having a higher risk of developing CPCH compared to males (*p* < 0.001). In addition, the presence of preoperative headaches significantly increased the risk of CPCH (*p* = 0.001). The occurrence of PCH is associated with younger age and female sex. In particular, female sex and preoperative headaches increased the risk of developing CPCH. These clinical factors should be considered in patients undergoing neurosurgery.

## 1. Introduction

Post-craniotomy headache (PCH) is an important postoperative complication that affects a significant proportion of patients undergoing craniotomy. Moderate to severe pain has been reported in up to 60–90% of patients undergoing craniotomy [1,2], with up to 30% of these patients progressing to a chronic phase of PCH, severely affecting their quality of life and disproportionately affecting vulnerable groups such as the elderly and frail [3]. In addition to its major impact on daily life after discharge from the hospital, PCH also affects in-hospital recovery [4]. Pain causes high blood pressure, which can lead to an increased risk of intracranial hemorrhage and intracranial hypertension. These complications not only prolong hospital stays but also increase mortality [5].

The International Headache Society defines PCH as a headache that occurs within 7 days of surgery and resolves within 3 months [6]. In some individuals, however, these headaches persist beyond this period and are classified as chronic post-craniotomy headaches (CPCHs). The development of CPCH is complex and influenced by a variety of factors, including surgical technique, craniotomy site, psychological predisposition, and especially the patient’s age, sex, and type of surgical approach [7,8,9]. Postoperative headache remains an important but unresolved problem in neurology and neurosurgery. Despite ongoing research by various groups worldwide, significant gaps remain in our understanding of the underlying mechanisms and risk factors for PCH and its progression to CPCH. In particular, while age and sex are known unmodified risk factors for primary and secondary headaches, their specific impact on PCH and its progression to chronicity in different surgical contexts has not been thoroughly investigated. The influence of age and sex on PCH has been the subject of conflicting evidence until recently [10], prompting the current study to address this knowledge gap.

Given the substantial burden of PCH and the need to understand its underlying mechanisms and risk factors, this study retrospectively analyzed data from five South Korean hospitals to identify the risk factors associated with the development of PCH and its progression to CPCH. While the roles of age and sex in headaches are known, their specific impact on PCH and CPCH in diverse surgical settings has not been extensively studied. Our research seeks to elucidate these relationships and provide a nuanced understanding of how demographic and surgical factors contribute to postoperative pain outcomes. Specifically, the association between demographic factors such as age and sex and pain intensity experienced during the first 7 days after surgery was examined. By identifying potential associations between age and sex and pain intensity in different surgical contexts, we aimed to clarify age- and sex-related differences in postoperative pain perception and pave the way for tailored pain management strategies based on patient demographics and craniotomy characteristics. Our goal is to improve patient outcomes, reduce the healthcare burden of PCH, a common postoperative complication, and ultimately improve the quality of life.

## 2. Materials and Methods

### 2.1. Data Collection

This study was a retrospective cohort study and collected clinical data from the Smart Clinical Data Warehouse (CDW) at Hallym University Medical Center (HUMC). The Smart CDW, based on the QlikView Elite solution (Qlik, Lund, Sweden), is used in the five hospitals of HUMC and provides analysis of text data from electronic medical records and integrated analysis of fixed data. The dataset included basic patient information, anesthesia and surgery records, laboratory test results, and prescription records from 1 January 2013 to 30 June 2023. This study was approved by the Institutional Review Board of Chuncheon Sacred Heart Hospital, Hallym University (IRB No. 2024-05-010). Because only deidentified data were used in this study, the IRB waived the requirement for informed consent from all subjects. 

### 2.2. Inclusion and Exclusion Criteria

The study included adult patients aged 18 years or older who underwent craniotomy or craniectomy between 1 January 2013 and 30 June 2023. This study excluded patients who underwent combined or complex surgery, patients who were unconscious for up to seven days after surgery, and patients who were lost to follow-up between seven days and three months after surgery. The flow chart for selecting a participant is shown in Figure 1.

### 2.3. Exposure and Primary and Secondary Outcomes 

The primary outcome of this study was defined as the worst headache intensity experienced by the patient during the first 7 days after surgery. Headache intensity was measured using the Numeric Rating Scale (NRS), which is expressed on a scale from 0 (no headache) to 10 (worst headache imaginable). Secondary outcomes were the progression from PCH to CPCH within three months of surgery and the association of several covariates with this progression. These covariates included demographic factors such as sex and preoperative variables, including the presence of preoperative headaches. In addition, the intensity of PCH within the first 7 days after surgery, as measured using the NRS, was also included as a covariate in the analysis.

### 2.4. Covariates 

Covariates included patient characteristics such as body mass index, comorbidities (hypertension, diabetes, headache, and sleep disorders), history of alcohol use, smoking status, and medications (opioids, sleeping pills, and psychotropic drugs), and perioperative factors such as the American Society of Anesthesiologists physical status, emergent status, type of surgery (craniotomy or craniectomy), and duration of surgery over 4 h.

### 2.5. Statistics 

We used the median and interquartile range (IQR) for continuous variables and frequencies and percentages for categorical variables. The Mann–Whitney test was used to compare continuous data between the PCH and control groups. The chi-squared test was used to compare categorical data. Multivariate analyses were used to assess factors associated with both the occurrence of PCH and its progression to CPCH. Odds ratios (ORs) and their 95% confidence intervals (CIs) were derived using logistic regression to quantify the risk of developing PCH and the subsequent risk of it becoming chronic. Fully adjusted ORs for both incident PCH and progression to CPCH were determined for each factor, including age and sex. To visualize the relationship between age and NRS, scatter plots with local polygon smoothing were used to generate smooth curves; this visualization method helps readers understand the data more clearly.

In this study, we examined the association between age and the worst headache experienced in the first 7 days after surgery using linear regression analysis, adjusting for covariates. We also calculated the beta coefficient and standard error for the relationship between age and NRS score. In a subgroup analysis, we examined the relationship between age and the worst headache in the first 7 days after surgery based on sex.

Bonferroni correction was used to assess statistical significance, and a *p*-value of less than 0.025 was considered significant. SPSS version 26.0 (IBM, Armonk, NY, USA) was used for data analysis, and R version 4.2.3 (R Foundation for Statistical Computing, https://www.r-project.org/ (accessed on 1 December 2023)) was used for graphs.

In this study, we calculated the statistical power for both age and sex variables using appropriate methods. For age, which is a continuous variable, we used the Mann–Whitney U test to compare median ages between groups. The effect size was estimated using Cliff’s delta (δ = 0.184), and the statistical power was subsequently determined to be >0.999 using the pwr package, with a Bonferroni-corrected significance level of 0.025. For sex, a binary variable, logistic regression was used to model the relationship between sex and the outcome of interest. The effect size was calculated from the odds ratio, and the power analysis was performed through simulation, resulting in a power of 0.988, also using the Bonferroni corrected significance level. These methods ensured robust power calculations, taking into account the specific characteristics of each variable. All analyses were performed using R, specifically utilizing the pwr, effsize, and stats packages for statistical modeling and power analysis.

## 3. Results

### 3.1. Patient Characteristics

This study initially enrolled 3923 patients aged 18 years or older who underwent craniotomy or craniectomy between January 2013 and June 2023. Of these, 2597 patients were excluded based on specific criteria, and 1326 patients were ultimately included in this study. The number, characteristics, and perioperative data of patients are summarized in detail in Table 1. 

### 3.2. Association of Age and Sex with the Risk of Post-Craniotomy Headache 

In the multivariate analysis for PCH, age was inversely associated with risk, indicating that younger age was associated with an increased risk of PCH (aOR: 0.98, 95% CI: 0.97–0.99, *p* = 0.003). In addition, being female showed a significant association with an increased risk of PCH (aOR: 1.56, 95% CI: 1.17–2.08, *p* = 0.002). These findings suggest that younger patients and females may require more attention for the potential development of PCH. (Table 2). 

### 3.3. Effects of Age and Sex on the Worst Headache up to 7 Days after Craniotomy 

The relationship between age and the severity of the most severe headache reported within the first 7 days after craniotomy is summarized in Table 3 and illustrated in Figure 2. Linear regression analysis shows a significant inverse relationship between age and the severity of the worst headache reported, with younger patients experiencing more severe headaches. The overall trend (β = −0.143, *p* < 0.001) is more pronounced when analyzed by gender, with younger female patients reporting greater severity of the worst headache (β = −0.178, *p* < 0.001) compared to their male counterparts (β = −0.142, *p* < 0.001). These results highlight that age, particularly younger age in females, is a significant factor in the intensity of the worst PCH experienced during the first week after surgery. Additionally, we analyzed the effect of drug categories on headache intensity in the treatment of PCH. Details of the categories and generic names of the drugs used for the treatment of PCH are given in Appendix A (Table A1 and Table A2).

### 3.4. Association of Age and Sex with the Risk of Chronic Post-Craniotomy Headaches 

Of the 927 patients who experienced PCH, 319 (34.4%) progressed to CPCH 3 months after surgery. The number, characteristics, and perioperative data of these patients are detailed in Table 4.

In the multivariable analysis evaluating risk factors for progression from PCH to CPCH, age showed a marginal association with decreased risk; each additional year slightly decreased the odds of PCH becoming chronic (aOR: 0.99, 95% CI: 0.98–1.00, *p* = 0.029). Gender was a significant factor, with females having a higher risk of developing CPCH compared to males (aOR: 1.99, 95% CI: 1.43–2.77, *p* < 0.001). In addition, the presence of preoperative headaches significantly increased the risk of CPCH, with these patients being more than twice as likely to develop chronic headaches (aOR: 2.65, 95% CI: 1.47–4.79, *p* = 0.001). The analysis also showed that the severity of the initial PCH, as assessed using the worst NRS within the first 7 days, was not a determinant of progression to CPCH, suggesting that initial pain intensity does not predict long-term outcomes (Table 5). 

## 4. Discussion

In this study using CDW over approximately 10 years, 69.9% of patients who underwent craniotomy or craniectomy experienced PCH. In the analysis of PCH occurrence, age was inversely related to risk, and female sex showed a significant association with an increased risk of PCH. There was also a significant inverse relationship between age and the severity of the worst headache, with younger female patients experiencing more severe headaches. In addition, 34.4% of patients who experienced PCH progressed to CPCH. Female sex and preoperative headaches increased the risk of developing CPCH. 

PCH is an under-recognized and very common adverse event following neurosurgery [11]. In a prospective study, the incidence of PCH in patients treated for intracranial aneurysms was 40%, with 10.7% of acute type and 29.3% of chronic type, according to the International Headache Society classification criteria [3]. A 5-year multicenter retrospective study reported that 49% of patients undergoing suboccipital craniotomy experienced PCH [12]. Another prospective observational study in China found that 53.3% of patients experienced post-craniotomy cervicogenic headaches after elective craniotomy [13]. In our study, 69.9% of patients who underwent craniotomy or craniectomy experienced PCH, and different rates of PCH have been reported depending on each surgical technique. Patients who underwent craniotomy had a significantly higher rate of headache than those who underwent craniectomy. There is some evidence that the incidence of PCH may vary according to the technique of neurosurgery. Previously reported patients undergoing elective craniotomy have a high rate of headache in over two-thirds of patients [14]. However, a study on headaches occurring after craniectomy found that 20–35% of patients who underwent vestibular neurinoma through a suboccipital lateral approach experienced post-craniectomy headache [15], which is lower than the incidence of PCH. The incidence of postoperative headache is higher in craniectomy than in craniotomy with bone flap replacement or cranioplasty [16,17,18,19]. Another multicenter retrospective study also found that craniectomy was a significant risk factor for PCH [12]. Compared with the results of other previous studies, the frequency of craniotomy was higher in the group of patients who developed PCH, and craniotomy was more associated with the development of PCH than craniectomy in our study. Further studies are needed to determine the relationship between the occurrence of PCH and neurosurgical techniques.

The pathogenesis of PCH is still unclear. The pathogenesis of PCH involves a complex interplay of several factors that contribute to the development and persistence of headaches after neurosurgery. Several multifactorial factors are known to be involved in PCH: mechanical factors through tissue disruption, vascular factors through changes in cerebral blood flow, neurogenic factors through trigeminal nerve activation and dural irritation, chemical mediators through the release of inflammatory molecules, preoperative factors through anesthesia-related headache and medication side effects, and individual susceptibility through genetic predisposition and pre-existing headache disorders [10,11,20,21,22,23,24,25]. 

In our study, a significant association was observed between age and sex and the risk of developing PCH. Younger age and female sex were identified as factors associated with an increased risk of PCH. The results highlighted an association between age, sex, and the severity of headaches experienced within the first seven days after surgery. Similarly, a previous study reported that age and sex were significantly associated with the onset of pain, with female and younger patients reporting higher percentages of postoperative pain [1]. Also, an age of <45 years (odds ratio = 3.0, *p* = 0.041) and an operation duration lasting > 4 h (odds ratio = 3.7, *p* = 0.019) were associated with the occurrence of acute PCH during the first week [26]. However, there is conflicting evidence as to whether age and sex may influence PCH [10,18,27,28]. Another study found no association between sex differences and PCH [27], which was supported by other studies [29,30]. However, female sex has also been reported to be associated with progression to CPCH [3,31]. In our analysis of patients with PCH who progressed to CPCH, factors such as sex and the presence of a preoperative headache significantly influenced the likelihood of developing CPCH, while age showed a marginal association with a reduced risk of CPCH. Preoperative anxiety may also predispose patients to greater postoperative pain after craniotomy [32]. In previous studies, women have higher levels of anxiety than men [33,34]. Sex differences exist at all levels of the signaling systems involved in pain processing and stress [35]. Several potential mechanisms have been proposed, including hormonal influence, pain perception, and psychosocial factors [36]. Fluctuating hormone levels during the menstrual cycle and hormone-related differences in cerebrovascular reactivity have long been implicated in the development of headache types [37,38,39]. Recent studies have identified sex differences in the physiological mechanisms underlying pain, including sex-specific involvement of multiple genes and proteins and differential interactions between hormones and the immune system that influence pain signaling. In addition, human neuroimaging studies have found sex and gender differences in pain-related neural circuits, including sex-specific brain changes in chronic pain processing [40]. Understanding these differences in pain perception based on age and sex can help optimize pain management strategies and support patients in the immediate postoperative period. Specifically, women were at increased risk for CPCH compared to men, highlighting the need for long-term surveillance of at-risk subgroups.

This multi-institutional retrospective study used comprehensive data collection, multivariate analyses, and rigorous statistical methods, contributing to the robustness of its findings. By identifying factors influencing the progression of PCH to CPCH, this study offers valuable insights for improving patient outcomes and reducing healthcare burdens related to these complications. 

However, this study has several limitations. Due to the retrospective design of this study, we collected clinical data from individuals presenting to a university medical center with five affiliated hospitals, and the cases collected did not meet the International Headache Society diagnostic criteria for headaches, which may limit the generalizability of our results to the general population. We also did not adjust for potential sources of selection bias and confounding variables. In addition, headaches were adjudicated based on medical record descriptions, and some patients did not provide a detailed description of their headaches. Unfortunately, we did not collect detailed clinical information on perioperative and postoperative factors, including the type of surgery, surgical repair technique, size of the bone flap, postoperative CSF leak, development of postoperative meningitis, and wound infection. Previous reports have shown that postoperative CSF sepsis, CSF leak, craniotomy size, and wound infection are important risk factors for PCH [18]. We also overlooked the psychological effects on PCH. It has been shown that psychological causes such as depression or anxiety disorder may be a predisposing factor for PCH and that these factors may be the result of pain conditions, indicating a bidirectional influence [41]. We did not analyze postoperative psychiatric factors using the anxiety and depression scale, although the premorbid use of psychotropic medications did not differ between the PCH and no PCH groups. Our study did not consider these clinical factors in its analysis, which may be a limitation. Future prospective, population-based studies are needed to investigate the association between various clinical factors and PCH. PCH is a condition that necessitates consideration of various clinical factors, underscoring the need for well-defined, standardized protocols in prospective studies. For instance, given existing studies that indicate a higher incidence of headaches in patients undergoing posterior fossa surgery, prospective research should incorporate standardized protocols regarding surgical sites, techniques, and treatment strategies.

## 5. Conclusions

In conclusion, this study sheds light on the multifaceted nature of PCH development, progression, and patient outcomes. By identifying key factors such as age, sex, and preoperative conditions that influence PCH and CPCH, the research paves the way for targeted interventions and personalized care strategies aimed at optimizing pain management and improving the long-term prognosis of individuals undergoing neurosurgery. Thus, recognition of the associations between age, sex, and PCH is essential for providing comprehensive and individualized care to patients undergoing neurosurgery.

## Figures and Tables

**Figure 1 biomedicines-12-01745-f001:**
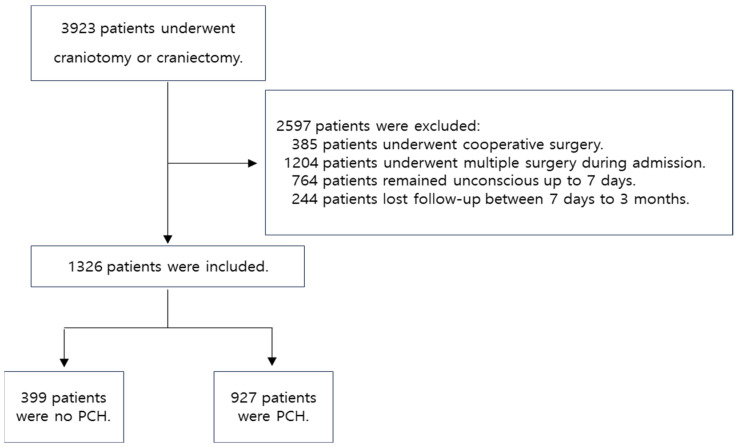
Flow chart of participant selection. PCH, post-craniotomy headache.

**Figure 2 biomedicines-12-01745-f002:**
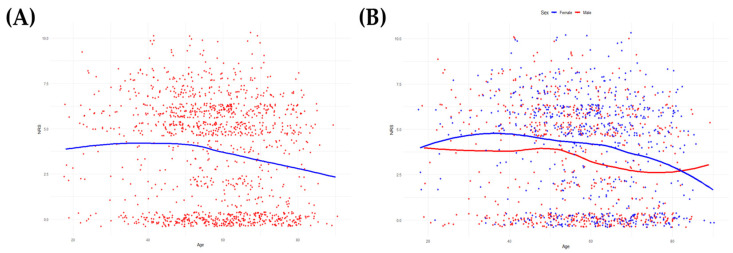
Scatterplot and locally estimated scatterplot smoothing line illustrating the correlation between age and the severity of post-craniotomy headache. (**A**) Trend for the total patient cohort. (**B**) Trend by sex, with blue indicating female patients and red indicating male patients.

**Table 1 biomedicines-12-01745-t001:** Characteristics and perioperative patient data for post-craniotomy headache patients.

PCH	No PCH	PCH	Total	*p*-Value
	(n = 399)	(n = 927)	(n = 1326)	
Age	61 (50–70)	58 (49–67)	58 (49–68)	0.001
Female	152 (38.1%)	466 (50.3%)	618 (46.6%)	<0.001
Body mass index	23.5 (20.9–25.6)	23.8 (21.5–26.0)	23.7 (21.3–25.9)	0.035
Hypertension	155 (38.8%)	330 (35.6%)	485 (36.6%)	0.287
Diabetes mellitus	60 (15.0%)	107 (11.5%)	167 (12.6%)	0.095
Headache	14 (3.5%)	53 (5.7%)	67 (5.1%)	0.122
Sleep disorder	17 (4.3%)	50 (5.4%)	67 (5.1%)	0.467
Alcohol Hx.	156 (39.1%)	329 (35.5%)	485 (36.6%)	0.235
Smoking Hx.	99 (24.8%)	241 (26.0%)	340 (25.6%)	0.700
Sleeping pills Hx.	9 (2.3%)	40 (4.3%)	49 (3.7%)	0.096
psychotropic drugs	7 (1.8%)	11 (1.2%)	18 (1.4%)	0.575
Emergency	286 (71.7%)	460 (49.6%)	746 (56.3%)	<0.001
Craniotomy	293 (73.4%)	824 (88.9%)	1117 (84.2%)	<0.001
Operation ≥ 4 h	157 (39.3%)	499 (53.8%)	656 (49.5%)	<0.001
ASA-PS ≥ 3	322 (80.7%)	532 (57.4%)	854 (64.4%)	<0.001
Opioid Hx.	3 (0.8%)	21 (2.3%)	24 (1.8%)	0.095

The data are presented in terms of medians and interquartile ranges or as numerical counts and corresponding percentages. PCH, post-craniotomy headache; ASA-PS, American Society of Anesthesiologists physical status; Hx., history; Opioid Hx., taking opioids for 30 days before hospitalization.

**Table 2 biomedicines-12-01745-t002:** Multivariate analysis of risk factors with the occurrence of post-craniotomy headaches.

PCH	Univariate	*p*-Value	Multivariate	*p*-Value
	OR (95% CI)		OR (95% CI)	
Age	0.99 (0.98–0.99)	0.001	0.98 (0.97–0.99)	0.003
Female	1.64 (1.29–2.09)	<0.001	1.56 (1.17–2.08)	0.002
Body mass index	1.04 (1.00–1.07)	0.035	1.02 (0.99–1.06)	0.249
Hypertension	0.87 (0.68–1.11)	0.260	1.10 (0.82–1.46)	0.530
Diabetes mellitus	0.74 (0.52–1.04)	0.079	0.84 (0.57–1.24)	0.393
Headache	1.67 (0.91–3.04)	0.095	1.17 (0.62–2.23)	0.622
Sleep disorder	1.28 (0.73–2.25)	0.389	0.80 (0.39–1.66)	0.554
Alcohol Hx.	0.86 (0.67–1.09)	0.211	0.99 (0.73–1.34)	0.950
Smoking Hx.	1.06 (0.81–1.40)	0.650	1.40 (1.01–1.95)	0.041
Sleeping pills Hx.	1.95 (0.94–4.07)	0.073	4.03 (1.47–11.00)	0.007
Psychotropic drugs	0.67 (0.26–1.75)	0.415	0.30 (0.10–0.93)	0.036
Emergency	0.39 (0.30–0.50)	<0.001	0.61 (0.45–0.83)	0.002
Craniotomy	2.89 (2.14–3.92)	<0.001	2.23 (1.61–3.09)	<0.001
Operation ≥ 4 h	1.80 (1.42–2.28)	<0.001	1.16 (0.88–1.52)	0.288
ASA-PS ≥ 3	0.32 (0.24–0.43)	<0.001	0.44 (0.33–0.60)	0.000
Opioid Hx.	3.06 (0.91–10.32)	0.071	2.81 (0.80–9.89)	0.107

The data are presented in terms of medians and interquartile ranges or as numerical counts and corresponding percentages. PCH, post-craniotomy headache; ASA-PS, American Society of Anesthesiologists physical status; Hx., history; Opioid Hx., taking opioids for 30 days before hospitalization.

**Table 3 biomedicines-12-01745-t003:** Linear regression of the relationship between age and the worst headache up to 7 days after craniotomy or craniectomy.

	R-Square	β ± SE	*p*-Value	30Yr
Total (n = 1326)	0.020	−0.143 ± 0.006	<0.001	4.3
Male (n = 708)	0.020	−0.142 ± 0.008	<0.001	4.3
Female (n = 618)	0.032	−0.178 ± 0.008	<0.001	5.3

**Table 4 biomedicines-12-01745-t004:** Characteristics and perioperative data for chronic post-craniotomy headache patients.

	No CPCH	CPCH	Total	*p*-Value
	(n = 608)	(n = 319)	(n = 927)	
Age	58 (49–68)	56 (48–65)	58 (49–67)	0.016
Female	266 (43.8%)	200 (62.7%)	466 (50.3%)	<0.001
Body mass index	23.7 (21.3–25.9)	23.9 (21.5–26.1)	23.8 (21.5–26.0)	0.344
Hypertension	228 (37.5%)	102 (32.0%)	330 (35.6%)	0.110
Diabetes mellitus	82 (13.5%)	25 (7.8%)	107 (11.5%)	0.014
Headache	22 (3.6%)	31 (9.7%)	53 (5.7%)	<0.001
Sleep disorder	33 (5.4%)	17 (5.3%)	50 (5.4%)	1.000
Alcohol Hx.	228 (37.5%)	101 (31.7%)	329 (35.5%)	0.091
Smoking Hx.	173 (28.5%)	68 (21.3%)	241 (26.0%)	0.023
Sleeping pills Hx.	27 (4.4%)	13 (4.1%)	40 (4.3%)	0.928
Psychotropic drugs	10 (1.6%)	1 (0.3%)	11 (1.2%)	0.145
Emergency	315 (51.8%)	145 (45.5%)	460 (49.6%)	0.077
Craniotomy	538 (88.5%)	286 (89.7%)	824 (88.9%)	0.669
Operation ≥ 4 h	302 (49.7%)	197 (61.8%)	499 (53.8%)	0.001
ASA-PS ≥ 3	375 (61.7%)	157 (49.2%)	532 (57.4%)	<0.001
Opioid Hx.	14 (2.3%)	7 (2.2%)	21 (2.3%)	1.000
NRS max	5 (4–6)	6 (5–7)	5 (5–6)	0.272
used drugs for treatment				
Opioids	211 (34.7%)	101 (31.7%)	312 (33.7%)	0.391
NSAIDs	421 (69.2%)	233 (73.0%)	654 (70.6%)	0.259
Antidepressants	109 (17.9%)	67 (21.0%)	176 (19.0%)	0.295
Anticonvulsants	553 (91.0%)	291 (91.2%)	844 (91.0%)	0.988
PCH				0.328
Mild (NRS 1–3)	106 (17.4%)	57 (17.9%)	163 (17.6%)	
Moderate (NRS 4–6)	374 (61.5%)	182 (57.1%)	556 (60.0%)	
Severe (NRS 7–10)	128 (21.1%)	80 (25.1%)	208 (22.4%)	

The data are presented in terms of medians and interquartile ranges or as numerical counts and corresponding percentages. CPCH, chronic post-craniotomy headache; PCH, post-craniotomy headache; ASA-PS, American Society of Anesthesiologists physical status; Hx., history; Opioid Hx., taking opioids for 30 days before hospitalization; NRS max, the maximal worst numeric rating scale.

**Table 5 biomedicines-12-01745-t005:** Multivariate analysis of risk factors with progression from post-craniotomy headache to chronic post-craniotomy headache.

	Univariate	*p*-Value	Multivariate	*p*-Value
	OR (95% CI)		OR (95% CI)	
Age	0.99 (0.98–1.00)	0.021	0.99 (0.98–1.00)	0.029
Female	2.16 (1.64–2.85)	<0.001	1.99 (1.43–2.77)	<0.001
Body mass index	1.02 (0.98–1.06)	0.344	1.01 (0.97–1.06)	0.502
Hypertension	0.78 (0.59–1.04)	0.095	0.98 (0.70–1.37)	0.914
Diabetes mellitus	0.55 (0.34–0.87)	0.011	0.62 (0.37–1.03)	0.064
Headache	2.87 (1.63–5.04)	<0.001	2.65 (1.47–4.79)	0.001
Sleep disorder	0.98 (0.54–1.79)	0.950	0.97 (0.43–2.18)	0.941
Alcohol Hx.	0.77 (0.58–1.03)	0.078	1.05 (0.73–1.49)	0.807
Smoking Hx.	0.68 (0.49–0.94)	0.019	0.82 (0.56–1.22)	0.331
Sleeping pills Hx.	0.91 (0.47–1.80)	0.795	1.18 (0.46–3.02)	0.725
psychotropic drugs	0.19 (0.02–1.48)	0.112	0.10 (0.01–0.89)	0.039
Emergency	0.78 (0.59–1.02)	0.066	1.11 (0.79–1.54)	0.556
Craniotomy	1.13 (0.73–1.75)	0.591	0.97 (0.61–1.56)	0.913
Operation ≥ 4 h	1.64 (1.24–2.16)	<0.001	1.37 (1.00–1.87)	0.048
ASA-PS ≥ 3	0.60 (0.46–0.79)	<0.001	0.71 (0.52–0.97)	0.032
Opioid Hx.	0.95 (0.38–2.38)	0.916	1.03 (0.38–2.79)	0.949
used drugs for treatment				
Opioids	0.87 (0.65–1.16)	0.352	0.76 (0.55–1.04)	0.088
NSAIDs	1.20 (0.89–1.63)	0.228	1.11 (0.80–1.54)	0.523
Antidepressants	1.22 (0.87–1.71)	0.257	1.28 (0.89–1.85)	0.186
Anticonvulsants	1.03 (0.64–1.66)	0.892	1.27 (0.75–2.14)	0.369
Mild (NRS 1–3)	Reference	0.329	Reference	0.348
Moderate (NRS 4–6)	0.90 (0.63–1.31)		0.86 (0.58–1.28)	
Severe (NRS 7–10)	1.16 (0.76–1.78)		1.11 (0.69–1.78)	

The data are presented in terms of medians and interquartile ranges or as numerical counts and corresponding percentages. ASA-PS, American Society of Anesthesiologists physical status; Hx., history; Opioid Hx., taking opioids for 30 days before hospitalization; NRS, numeric rating scale.

## Data Availability

Data are contained within the article.

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
