# Peer review of "Influence of Age and Sex on Post-Craniotomy Headache"

_biomedicines, 2024, doi:10.3390/biomedicines12081745_

Round 1
Reviewer 1 Report
Comments and Suggestions for Authors
The topic of postoperative headache is an important but still unresolved problem in neurology and neurosurgery. This problem is being actively studied by various research groups around the world, so I recommend that the authors expand the Introduction section and explain what previously unresolved issues prompted them to conduct this study. The age and sex of patients are well-known unmodified risk factors for primary and secondary headaches. What is the scientific novelty of this study?
Please move the flowchart of this study to the Materials and Methods section.
What method was used by the authors to calculate the sample size?
Have the authors evaluated the technique of surgical intervention, the development of infection, postoperative leakage of cerebrospinal fluid, postoperative meningitis, bone flap size and wound infection?
The authors used only a Numerical Rating Scale to assess the intensity of postoperative headache. However, postoperative anxiety and depression may be known modifiable factors of postoperative headache. An important and simple tool to address this problem is the Hospital Anxiety and Depression Scale.
For the treatment of postoperative (postcraniotomy) headache, not only non-steroidal anti-inflammatory headaches are widely used, but also antidepressants and anti-convulsants, since the use of opioid analgesics, although very effective in the short term, should be limited, given the rapid increase in the number of opioid dependence patients worldwide.
What drugs were used to correct postoperative headache in the observed patients? How did this affect the intensity of headache in male and female patients?
In addition, 18 of the 37 articles cited by the authors were published more than 10 years ago. Since this topic has been actively studied over the past 5-10 years, I strongly recommend that the authors modify the Introduction and Discussion sections taking into account the research results of recent years.
Author Response
July 20, 2024
Reviewer 1
Biomedicine
Dear Reviewer 1,
Please find attached a revised version of our manuscript, “Influence of age and sex on post-craniotomy headache” (biomedicine-3108820).
We thank you for your thoughtful suggestions regarding the original version of our paper; most of the suggested changes have been incorporated into the revision.
All revisions are described in detail in the order mentioned in the review, following your comments (in italics). We believe that the revisions have greatly improved the manuscript and hereby submit the revised version for publication.
Comments to author:
The topic of postoperative headache is an important but still unresolved problem in neurology and neurosurgery. This problem is being actively studied by various research groups around the world, so I recommend that the authors expand the Introduction section and explain what previously unresolved issues prompted them to conduct this study. The age and sex of patients are well-known unmodified risk factors for primary and secondary headaches. What is the scientific novelty of this study?
We thank the reviewer for their comments and suggestions, that helped us improve our manuscript.
We appreciate your suggestions for improving the Introduction section. We recognize the importance of addressing on the unresolved issues that prompted our study. To address this, we will expand the Introduction to include a discussion of the gaps in current research. Specifically, while age and sex are recognized as risk factors for primary and secondary headache, there is limited understanding of how these factors interact with surgical variables to influence the development and persistence of post-craniotomy headache (PCH) and chronic post-craniotomy headache (CPCH). Our study aims to fill this gap by analyzing demographic factors in relation to surgical techniques and contexts.
The scientific novelty of our study lies in its comprehensive approach to identify specific risk factors associated with the progression of PCH to CPCH, particularly in the context of South Korean hospitals. While the roles of age and sex in headache is well established, their specific impact on PCH and CPCH in different surgical settings has not been extensively studied. Our research seeks to elucidate these relationships and provide a nuanced understanding of how demographic and surgical factors contribute to postoperative pain outcomes. This includes examining the intensity of pain experienced during the first 7 days after surgery and identifying potential age- and sex-related differences in pain perception, that may inform tailored pain management strategies.
Therefore, we have added and revised the text in the Introduction as follows:
Postoperative headache remains an important but unresolved problem in neurology and neurosurgery. Despite ongoing research by various groups worldwide, significant gaps remain in our understanding of the underlying mechanisms and risk factors for PCH and its progression to CPCH. In particular, while age and sex are known unmodified risk fac-tors for primary and secondary headache, their specific impact on PCH and its progression to chronicity in different surgical contexts has not been thoroughly investigated. The influence of age and sex on PCH has been the subject of conflicting evidence until recently [10], prompting the current study to address this knowledge gap.
(page 2, lines 52 – page 2, lines 59)
While the roles of age and sex in headache are known, their specific impact on PCH and CPCH in diverse surgical settings has not been extensively studied. Our research seeks to elucidate these relationships and provide a nuanced understanding of how demographic and surgical factors contribute to postoperative pain outcomes.
(page 2, lines 63 – page 2, lines 66)
Please move the flowchart of this study to the Materials and Methods section.
Thank you. We have moved Figure 1 and the corresponding contents from the Results section to the Methods section.
The flow chart for selecting a participant is shown in Figure 1 (Figure 1).
(page 2, lines 93 – page 2, lines 94)
What method was used by the authors to calculate the sample size?
Thank you. In retrospective studies, where the analysis is based on data that has already been collected, it is generally more appropriate to calculate the power of the study. Because retrospective studies are based on data that have already been collected, it is not possible to add new samples or change the sample size. Therefore, it seems more appropriate to assess the statistical power of the research results using the existing data. We have added text to the Methods section as follows:
In this study, we calculated the statistical power for both age and sex variables using appropriate methods. For age, which is a continuous variable, we used the Mann-Whitney U test to compare median ages between groups. The effect size was estimated using Cliff's delta (δ = 0.184), and statistical power was subsequently determined to be >0.999 using the pwr package, with a Bonferroni-corrected significance level of 0.025. For sex, a binary variable, logistic regression was used to model the relationship between sex and the outcome of interest. The effect size was calculated from the odds ratio, and the power analysis was performed by simulation, resulting in a power of 0.988, also using the Bonferroni corrected significance level. These methods ensured robust power calculations, taking into account the specific characteristics of each variable. All analyses were performed using R, specifically utilizing the pwr, effsize, and stats packages for statistical modeling and power analysis.
(page 4, lines 139 – page 4, lines 150)
Have the authors evaluated the technique of surgical intervention, the development of infection, postoperative leakage of cerebrospinal fluid, postoperative meningitis, bone flap size and wound infection?
Thank you. Our study design was retrospective, and unfortunately, we did not collect accurate clinical information regarding the type of surgery, surgical repair technique, bone flap size, postoperative CSF leak, and development of postoperative infection, including postoperative meningitis, wound infection. Previous reports have shown that postoperative CSF sepsis, CSF leak, craniotomy size, and wound infection are important risk factors for PCH. Our study did not consider these peri- and postoperative factors in its analysis, which may be a limitation of the study. Therefore, we added this as a limitation in the Discussion as follows:
Unfortunately, we did not collect detailed clinical information on perioperative and postoperative factors, including the type of surgery, surgical repair technique, size of bone flap, postoperative CSF leak, development of postoperative meningitis, and wound infection. Previous reports have shown that postoperative CSF sepsis, CSF leak, craniotomy size, and wound infection are important risk factors for PCH.
(page 12, lines 315 – page 12, lines 319)
Our study did not consider these clinical factors in its analysis, which may be a limitation.
(page 12, lines 325 – page 12, lines 326)
We also added the following citation.
- Sezer C.; Gokten, M. ; Sezer, A.; Gezgin, I.; Binboga, A.I.; Onay, M. Role of Craniectomy Versus Craniotomy via the Retrosigmoid Approach in Decreasing the Incidence of Postoperative Headache. Int Surg 2022, 106 (1), 32–38, https://doi.org/10.9738/INTSURG-D-21-00011.1
The authors used only a Numerical Rating Scale to assess the intensity of postoperative headache. However, postoperative anxiety and depression may be known modifiable factors of postoperative headache. An important and simple tool to address this problem is the Hospital Anxiety and Depression Scale.
Thank you. Psychological disorders such as anxiety and depression can contribute to chronic pain. Although it seems clear that anxiety and depression can be a consequence of the pain condition, the effect is bidirectional. Postoperative anxiety and depression may be known modifiable factors of PCH. We analyzed the use of psychotropic medications, but there was no difference between the PCH group and the PCH-free group. However, we did not analyze postoperative psychiatric factors using the anxiety and depression scale. This may be a limitation of our study, and we added it to the limitations in the discussion section as follows:
We also overlooked the psychological effects on PCH. It has been shown that psychological causes such as depression or anxiety disorder may be a predisposing factor for PCH, and that these factors may be the result of pain conditions, indicating a bidirectional influence [43]. We did not analyze postoperative psychiatric factors using the anxiety and depression scale, although the premorbid use of psychotropic medications did not differ between the PCH and no PCH groups. Our study did not consider these clinical factors in its analysis, which may be a limitation.
(page 12, lines 319 – page 12, lines 326)
We also added the following citation.
- Thomas, M.; Rampp, S.; Scheer, M.; Strauss, C.; Prell, J.; Schönfeld, R.; Leplow, B. Premorbid Psychological Factors Associated with Long-Term Postoperative Headache after Microsurgery in Vestibular Schwannoma—A Retrospective Pilot Study. Brain Sci. 2023, 13(8), 1171, https://doi.org/10.3390/brainsci13081171.
For the treatment of postoperative (postcraniotomy) headache, not only non-steroidal anti-inflammatory headaches are widely used, but also antidepressants and anti-convulsants, since the use of opioid analgesics, although very effective in the short term, should be limited, given the rapid increase in the number of opioid dependence patients worldwide. What drugs were used to correct postoperative headache in the observed patients? How did this affect the intensity of headache in male and female patients?
Thank you. For the treatment of post-craniotomy headache (PCH), we have administered medications including opioids, NSAIDs, antidepressants, and anticonvulsants. Details of the categories and generic names of the medications used are provided in Appendix 1. In addition, we have reanalyzed the effects of different types of medications on PCH. Tables 4 and 5 have been revised to examine the effects by sex. In addition, an analysis of the effect of medication categories on headache intensity has been included in Appendix 2.
We have added the text in the Results section as follows:
Additionally, we analyzed the effect of drug categories on headache intensity in the treatment of PCH. Details of the categories and generic names of the drugs used for treatment of PCH are given in the Appendix (Table A 1&2).
(page 6, lines 187 – page 6, lines 190)
We have revised the text in Table 4.5 and added tables to the Appendix as follows:
Table 4. Linear regression of the relationship between sex and the worst headache up to 7 days after craniotomy or craniectomy.
|
No CPCH |
CPCH |
Total |
p |
(N=608) |
(N=319) |
(N=927) |
||
Age |
58 (49-68) |
56 (48-65) |
58 (49-67) |
0.016 |
Female |
266 (43.8%) |
200 (62.7%) |
466 (50.3%) |
< 0.001 |
Body mass index |
23.7 (21.3-25.9) |
23.9 (21.5-26.1) |
23.8 (21.5-26.0) |
0.344 |
Hypertension |
228 (37.5%) |
102 (32.0%) |
330 (35.6%) |
0.110 |
Diabetes mellitus |
82 (13.5%) |
25 (7.8%) |
107 (11.5%) |
0.014 |
Headache |
22 (3.6%) |
31 (9.7%) |
53 (5.7%) |
< 0.001 |
Sleep disorder |
33 (5.4%) |
17 (5.3%) |
50 (5.4%) |
1.000 |
Alcohol Hx. |
228 (37.5%) |
101 (31.7%) |
329 (35.5%) |
0.091 |
Smoking Hx. |
173 (28.5%) |
68 (21.3%) |
241 (26.0%) |
0.023 |
Sleeping pills Hx. |
27 (4.4%) |
13 (4.1%) |
40 (4.3%) |
0.928 |
psychotropic drugs |
10 (1.6%) |
1 (0.3%) |
11 (1.2%) |
0.145 |
Emergency |
315 (51.8%) |
145 (45.5%) |
460 (49.6%) |
0.077 |
Craniotomy |
538 (88.5%) |
286 (89.7%) |
824 (88.9%) |
0.669 |
Operation ≥ 4hr |
302 (49.7%) |
197 (61.8%) |
499 (53.8%) |
0.001 |
ASA-PS ≥ 3 |
375 (61.7%) |
157 (49.2%) |
532 (57.4%) |
< 0.001 |
Opioid Hx. |
14 (2.3%) |
7 (2.2%) |
21 (2.3%) |
1.000 |
NRS max |
5 (4-6) |
6 (5-7) |
5 (5-6) |
0.272 |
used drugs for treatment |
||||
Opioids |
211 (34.7%) |
101 (31.7%) |
312 (33.7%) |
0.391 |
NSAIDs |
421 (69.2%) |
233 (73.0%) |
654 (70.6%) |
0.259 |
Antidepressants |
109 (17.9%) |
67 (21.0%) |
176 (19.0%) |
0.295 |
Anticonvulsants |
553 (91.0%) |
291 (91.2%) |
844 (91.0%) |
0.988 |
PCH |
|
|
|
0.328 |
Mild (NRS 1-3) |
106 (17.4%) |
57 (17.9%) |
163 (17.6%) |
|
Moderate (NRS 4-6) |
374 (61.5%) |
182 (57.1%) |
556 (60.0%) |
|
Severe (NRS 7-10) |
128 (21.1%) |
80 (25.1%) |
208 (22.4%) |
|
Table 5. Multivariate analysis of risk factors with progression from post-craniotomy headache to chronic post-craniotomy headache.
|
Univariate |
p |
Multivariate |
p |
OR (95% CI) |
OR (95% CI) |
|||
Age |
0.99 (0.98-1.00) |
0.021 |
0.99 (0.98-1.00) |
0.029 |
Female |
2.16 (1.64-2.85) |
< 0.001 |
1.99 (1.43-2.77) |
< 0.001 |
Body mass index |
1.02 (0.98-1.06) |
0.344 |
1.01 (0.97-1.06) |
0.502 |
Hypertension |
0.78 (0.59-1.04) |
0.095 |
0.98 (0.70-1.37) |
0.914 |
Diabetes mellitus |
0.55 (0.34-0.87) |
0.011 |
0.62 (0.37-1.03) |
0.064 |
Headache |
2.87 (1.63-5.04) |
< 0.001 |
2.65 (1.47-4.79) |
0.001 |
Sleep disorder |
0.98 (0.54-1.79) |
0.950 |
0.97 (0.43-2.18) |
0.941 |
Alcohol Hx. |
0.77 (0.58-1.03) |
0.078 |
1.05 (0.73-1.49) |
0.807 |
Smoking Hx. |
0.68 (0.49-0.94) |
0.019 |
0.82 (0.56-1.22) |
0.331 |
Sleeping pills Hx. |
0.91 (0.47-1.80) |
0.795 |
1.18 (0.46-3.02) |
0.725 |
psychotropic drugs |
0.19 (0.02-1.48) |
0.112 |
0.10 (0.01-0.89) |
0.039 |
Emergency |
0.78 (0.59-1.02) |
0.066 |
1.11 (0.79-1.54) |
0.556 |
Craniotomy |
1.13 (0.73-1.75) |
0.591 |
0.97 (0.61-1.56) |
0.913 |
Operation ≥ 4hr |
1.64 (1.24-2.16) |
< 0.001 |
1.37 (1.00-1.87) |
0.048 |
ASA-PS ≥ 3 |
0.60 (0.46-0.79) |
< 0.001 |
0.71 (0.52-0.97) |
0.032 |
Opioid Hx. |
0.95 (0.38-2.38) |
0.916 |
1.03 (0.38-2.79) |
0.949 |
used drugs for treatment |
||||
Opioids |
0.87 (0.65-1.16) |
0.352 |
0.76 (0.55-1.04) |
0.088 |
NSAIDs |
1.20 (0.89-1.63) |
0.228 |
1.11 (0.80-1.54) |
0.523 |
Antidepressants |
1.22 (0.87-1.71) |
0.257 |
1.28 (0.89-1.85) |
0.186 |
Anticonvulsants |
1.03 (0.64-1.66) |
0.892 |
1.27 (0.75-2.14) |
0.369 |
Mild (NRS 1-3) |
Reference |
0.329 |
Reference |
0.348 |
Moderate (NRS 4-6) |
0.90 (0.63-1.31) |
|
0.86 (0.58-1.28) |
|
Severe (NRS 7-10) |
1.16 (0.76-1.78) |
|
1.11 (0.69-1.78) |
|
Appendix
Table A1. Effect of drug categories used on headache intensity in the treatment of PCH
|
Mild |
Moderate (N=556) |
Severe (N=208) |
Total (N=927) |
p |
Opioids |
37 (22.7%) |
166 (29.9%) |
109 (52.4%) |
312 (33.7%) |
< 0.001 |
NSAIDs |
121 (74.2%) |
391 (70.3%) |
142 (68.3%) |
654 (70.6%) |
0.450 |
antidepressants |
21 (12.9%) |
101 (18.2%) |
54 (26.0%) |
176 (19.0%) |
0.005 |
anticonvulsants |
154 (94.5%) |
508 (91.4%) |
182 (87.5%) |
844 (91.0%) |
0.060 |
Table A 2. Categories and generic names of drug used to treat PCH
Categories |
Generic names of drugs used |
Opioids |
Codeine, Fentanyl, Hydromorphone, Morphine, Oxycodone, Pethidine, Tapentadol |
NSAIDs |
Aceclofenac, Celecoxib, Dexibuprofen, Diclofenac, Etoricoxib, Ibuprofen, Ketoprofen, Ketorolac, Lornoxicam, Loxoprofen, Mefenamic, Morniflumate, Nabumetone, Naproxen, Pelubiprofen, Proglumetacin maleate, Propacetamol, Talniflumate, Zaltoprofen |
Antidepressants |
Agomelatine, Amitriptyline, Bupropion, Duloxetine, Escitalopram, Escitalopram oxalate, Fluoxetine, Milnacipran, Mirtazapine, Nortriptyline, Paroxetine, Sertraline, Tianeptine, Trazodone, Venlafaxine, Vortioxetine hydrobromide |
Anticonvulsants |
Carbamazepine, Clonazepam, Diphenylhydantoin, Divalproex, Divalproex sodium, Fosphenytoin, Gabapentin, Lacosamide, Lamotrigine, Levetiracetam, Perampanel, Pregabalin, Sodium Valproate, Topiramate |
PCH, post-craniotomy headache.
In addition, 18 of the 37 articles cited by the authors were published more than 10 years ago. Since this topic has been actively studied over the past 5-10 years, I strongly recommend that the authors modify the Introduction and Discussion sections taking into account the research results of recent years.
Thank you. We have revised the text in the Introduction and Discussion sections as follows:
In addition to its major impact on daily life after discharge from the hospital, PCH also affects in-hospital recovery. Pain causes high blood pressure, which can lead to an increased risk of intracranial hemorrhage and intracranial hypertension. These complications not only prolong hospital stay, but also increases mortality.
(page 1, lines 39 – page 1, lines 43)
The influence of age and sex on PCH has been the subject of conflicting evidence until recently, prompting the current study to address this knowledge gap.
(page 2, lines 57 – page 2, lines 59)
Several potential mechanisms have been proposed, including hormonal influence, pain perception, and psychosocial factors. Fluctuating hormone levels during the menstrual cycle and hormone-related differences in cerebrovascular reactivity have long been implicated in the development of headache types.
(page 11, lines 282 – page 11, lines 285)
We have also changed and added more recent citations as follows:
- Basali A, Mascha EJ, Kalfas I, Schubert A. Relation between perioperative hypertension and intracranial hemorrhage after craniotomy. Anesthesiology. 2000, 93, 48-54. DOI: 10.1097/00000542-200007000-00012.
- Chowdhury, T.; Garg, R.; Sheshadri, V.; Venkatraghavan, L.; Bergese, S.D.; Cappellani, R.B.;
- Habibi,M.A.,; Kobets, A.J.; Boskabadi, A.R.; Nasab, M.M.; Sobhanian, P.; Hamishegi, F.S.; Alavi, S.A.N., A comprehensive systematic review and meta‑analysis study in comparing decompressive craniectomy versus craniotomy in patients with acute subdural hematoma. Neurosurg Review 2024 47,77, https://doi.org/10.1007/s10143-024-02292-5.
- Azouz,H.M.; Sofar, H.M.H.; Abbass, W.A.; Ahmed, Ahmed. E.; El‑far, M.T., Comparative study between the outcome of decompressive craniotomy versus craniectomy in the management of acute subdural hematoma. Egyptian J Neurosurg 2023, 38,30, https://doi.org/10.1186/s41984-023-00209-w.
- Sezer C.; Gokten, M. ; Sezer, A.; Gezgin, I.; Binboga, A.I.; Onay, M. Role of Craniectomy Versus Craniotomy via the Retrosigmoid Approach in Decreasing the Incidence of Postoperative Headache. Int Surg 2022, 106 (1), 32–38, https://doi.org/10.9738/INTSURG-D-21-00011.1.
- Lutman, B.; Bloom, J.; Nussenblatt, B..; Romo V., A Contemporary Perspective on the Management of Post-Craniotomy Headache and Pain. Curr Pain Headache Rep 2018 22: 69, https://doi.org/10.1007/s11916-018-0722-4.
- Haldar R, Kaushal A, Gupta D, Srivastava S, Singh PK. Pain following craniotomy: reassessment of the available options. Biomed Res Int. 2015;2015:509164. https://doi.org/10.1155/2015/509164.
- Samagh, N.; Jangra, K.; Dey, A., Post-craniotomy Pain: An Update. J Neuroanaesth Crit Care 2023, 10(01), 021-030, DOI: 10.1055/s-0042-1760271.
- Bello, C.; Andereggen, L.; Luedi, M.M.; Beilstein, C.M., Postcraniotomy Headache: Etiologies and Treatments. Curr Pain Headache Rep 2022, 26, 357–364, https://doi.org/10.1007/s11916-022-01036-8.
- Phoowanakulchai, S.; Ida, M.; Naito, Y.; Kawaguchi, M., Persistent incisional pain at 1 year after craniotomy: a retro-spective observational study. BMC Anesthesiol 2023, 23, 115, https://doi.org/10.1186/s12871-023-02068-2.
- Hipolito Rodrigues MA, Maitrot-Mantelet L, Plu-Bureau G, Gompel A. Migraine, hormones and the menopausal tran-sition. Climacteric. 2018. 21.256–66. doi: 10.1080/13697137.2018.1439914.
- Thomas, M.; Rampp, S.; Scheer, M.; Strauss, C.; Prell, J.; Schönfeld, R.; Leplow, B. Premorbid Psychological Factors Associated with Long-Term Postoperative Headache after Microsurgery in Vestibular Schwannoma—A Retrospective Pilot Study. Brain Sci. 2023, 13(8), 1171, https://doi.org/10.3390/brainsci13081171.
We have addressed all of the issues raised by the reviewers. We are grateful for the constructive comments made during the review process. We believe that our paper has been improved by implementing these suggestions.
Yours faithfully,
Jong-Hee Sohn, M.D. Ph.D.
Department of Neurology, Chuncheon Sacred Heart Hospital, Hallym University College of Medicine, 77 Sakju-ro, Chuncheon-si, Gangwon-state, 24253, Republic of Korea
Tel: +82-33-252-9970, Fax: +82-33-241-8063
E-mail: deepfoci@hallym.or.kr
Jae-June Lee M.D. Ph.D.
Departments of Anesthesiology and Pain medicine, Chuncheon Sacred Heart Hospital, Hallym University College of Medicine, 77 Sakju-ro, Chuncheon-si, Gangwon-state, 24253, Republic of Korea
Tel: +82-33-252-9970, Fax: +82-33-241-8063
E-mail: iloveu59@hallym.or.kr

Reviewer 2 Report
Comments and Suggestions for Authors
This retrospective cohort study is interesting and seeks to highlight the possible clinical variables recorded in craniotomy patients with post-operative headache.
Although somewhat different case studies are cited in the literature, the data reported are interesting and the case studies numerous enough to be convincing. The job description is correct, but the discussion session is too long and should be shortened to facilitate reading of the text and its understanding. The limitations of the research are well exposed, and the non-application of international definition criteria for headaches partially limits the generalization of the data collected.
The conclusions are consistent with the data obtained
Author Response
July 20, 2024
Reviewer 2
Biomedicine
Dear Reviewer 2,
Please find attached a revised version of our manuscript, “Influence of age and sex on post-craniotomy headache” (biomedicine-3108820).
We thank you for your thoughtful suggestions regarding the original version of our paper; most of the suggested changes have been incorporated into the revision.
All revisions are described in detail in the order mentioned in the review, following your comments (in italics). We believe that the revisions have greatly improved the manuscript and hereby submit the revised version for publication.
Comments to author:
This retrospective cohort study is interesting and seeks to highlight the possible clinical variables recorded in craniotomy patients with post-operative headache.
Although somewhat different case studies are cited in the literature, the data reported are interesting and the case studies numerous enough to be convincing. The job description is correct, but the discussion session is too long and should be shortened to facilitate reading of the text and its understanding. The limitations of the research are well exposed, and the non-application of international definition criteria for headaches partially limits the generalization of the data collected. The conclusions are consistent with the data obtained.
We thank the reviewer for their comments and suggestions, which have helped us improve our manuscript.
Following the reviewer's recommendation, we have revised the text in the Discussion as follows:
This multi-institutional retrospective study used comprehensive data collection, multivariate analyses, and rigorous statistical methods, contributing to the robustness of its findings. By identifying factors influencing the progression of PCH to CPCH, the study offers valuable insights for improving patient outcomes and reducing healthcare burdens related to these complications.
(page 11, lines 295 – page 11, lines 299)
Due to the retrospective design of this study, we collected clinical data from individuals presenting to a university medical center with five affiliated hospitals, and the cases collected did not meet the International Headache Society diagnostic criteria for headache, which may limit the generalizability of our results to the general population.
(page 11, lines 308 – page 11 , lines 312)
We have addressed all of the issues raised by the reviewers. We are grateful for the constructive comments made during the review process. We believe that our paper has been improved by implementing these suggestions.
Yours faithfully,
Jong-Hee Sohn, M.D. Ph.D.
Department of Neurology, Chuncheon Sacred Heart Hospital, Hallym University College of Medicine, 77 Sakju-ro, Chuncheon-si, Gangwon-state, 24253, Republic of Korea
Tel: +82-33-252-9970, Fax: +82-33-241-8063
E-mail: deepfoci@hallym.or.kr
Jae-June Lee M.D. Ph.D.
Departments of Anesthesiology and Pain medicine, Chuncheon Sacred Heart Hospital, Hallym University College of Medicine, 77 Sakju-ro, Chuncheon-si, Gangwon-state, 24253, Republic of Korea
Tel: +82-33-252-9970, Fax: +82-33-241-8063
E-mail: iloveu59@hallym.or.kr

Reviewer 3 Report
Comments and Suggestions for Authors
For Authors:
The purpose of this retrospective cohort study and collect clinical data of adult patients who underwent cranial surgery over 10 years (2013-2023) at 5 hospitals in Korea that are all part of the Hallym University Medical Center (HUMC). The purpose was to find a way to be able to understand which of these patients would have a high likelihood of having severe headache as an effect of undergoing this surgery. The hope was to improve long term outcoms for individuals having this surgery. During the 10 years 3,923 underwent this surgery. Of that number 1,326 were found to have significant information both of the effect in hospital and later. The difference was due to exclusions issues of multiple surgeries in one admission, prolonged unconsciousness and failure of follow up.
Analysis of these data would lead to awareness of the problems of headache and hopefully in prevent or recognize these problems to decrease the problems for the patient. In the Introduction the authors state that these headaches in the postoperative time may become prolonged. They also say that severe pain in the postoperative time may lead to increased mortality and healthcare cost. For these issues they give references 4 and 12. In reading 4 I found nothing about the mortality or cost. I was not able to access number 12. Knowledge of causation of pain as shown by imaging could lead to improved treatment as to the value of specific imaging The most important aspect discussed here relates to the importance of the family to what has happened, what to expect, and where to go when there are headaches. The study states that it is often difficult to have access to neurosurgery for patients with headaches.
A total of 16 characteristics were subject to statistic evaluation both in a univariate and multivariant analysis. Five of the 16 were found to have significant difference including being female, having emergency for surgery, Craniotomy and not craniectomy, operation taking over 4 hours and QSA-PS outcome measurement. The most important issues related to Female sex and younger age. The latter did not seem to be significant on table 4.
The data base used in this study is likely to have the reason for the surgery such as trauma, tumor or aneurysm and what part of the calvarium is opened. Including these factors would increase the value of the study.
One of the important characters was that there was a history of headache prior to the surgery. The outcomes here are to be expected. It is known that young females prior to menopause have much higher problems with headaches especially of migraine which tends to go away with menopause. I am not certain that knowing who is likely to have the headaches may lead to different forms of treatment and plans for treatment with medication or treatment of activity etcetera. Shouldn’t there be a plan for management for all patients?
The references are important with studies of a number of different reasons for surgery. From some of the references it seems as if the highest cause of headache occurs in those patients undergoing surgery of the posterior fossa. From the references it is clear that there is not an accepted protocol for post craniotomy pain. It might make the study more interesting if the authors would put a brief discussion of the next step to a protocol.
Author Response
July 20, 2024
Reviewer 3
Biomedicine
Dear Reviewer 3,
Please find attached a revised version of our manuscript, “Influence of age and sex on post-craniotomy headache” (biomedicine-3108820).
We thank you for your thoughtful suggestions regarding the original version of our paper; most of the suggested changes have been incorporated into the revision.
All revisions are described in detail in the order mentioned in the review, following your comments (in italics). We believe that the revisions have greatly improved the manuscript and hereby submit the revised version for publication.
Comments to author:
The purpose of this retrospective cohort study and collect clinical data of adult patients who underwent cranial surgery over 10 years (2013-2023) at 5 hospitals in Korea that are all part of the Hallym University Medical Center (HUMC). The purpose was to find a way to be able to understand which of these patients would have a high likelihood of having severe headache as an effect of undergoing this surgery. The hope was to improve long term outcomes for individuals having this surgery. During the 10 years 3,923 underwent this surgery. Of that number 1,326 were found to have significant information both of the effect in hospital and later. The difference was due to exclusions issues of multiple surgeries in one admission, prolonged unconsciousness and failure of follow up.
Analysis of these data would lead to awareness of the problems of headache and hopefully in prevent or recognize these problems to decrease the problems for the patient. In the Introduction the authors state that these headaches in the postoperative time may become prolonged. They also say that severe pain in the postoperative time may lead to increased mortality and healthcare cost.
We thank the reviewer for their comments and suggestions, which have helped us improve our manuscript.
For these issues they give references 4 and 12. In reading 4, I found nothing about the mortality or cost. I was not able to access number 12. Knowledge of causation of pain as shown by imaging could lead to improved treatment as to the value of specific imaging. The most important aspect discussed here relates to the importance of the family to what has happened, what to expect, and where to go when there are headaches. The study states that it is often difficult to have access to neurosurgery for patients with headaches.
Thank you. We have revised the text in the Introduction and Discussion section as follows:
In addition to its major impact on daily life after discharge from the hospital, PCH also affects in-hospital recovery [4]. Pain causes high blood pressure, which can lead to an increased risk of intracranial hemorrhage and intracranial hypertension. These complications not only prolong hospital stay, but also increases mortality [5].
(page 1, lines 39 – page 1, lines 43)
Previously reported patients undergoing elective craniotomy have a high rate of headache in over two-thirds of patients [14].
(page 10, lines 245 – page 10, lines 246)
We have also changed and added the citations as follows:
- Leslie, K.; Troedel, S.; Irwin, K.; Pearce, F.; Ugoni, A.; Gillies, R.; Pemberton, E.; Dharmage, S. Quality of recovery from anesthesia in neurosurgical patients. Anesthesiology 2003, 99, 1158-1165, doi:10.1097/00000542-200311000-00024.
- Basali A, Mascha EJ, Kalfas I, Schubert A. Relation between perioperative hypertension and intracranial hemorrhage after craniotomy. Anesthesiology. 2000, 93, 48-54. DOI: 10.1097/00000542-200007000-00012.
- Sriganesh, K.; Kramer, B.W.; Wadhwa, A.; Akash, V.S.; Bharadwaj, S.; Rao, G.S.U.; Steinbusch, H.W.M.; Konar, S.K.; Gopalakrishna, K.N.; Sathyaprabha, T.N., Incidence, predictors, and impact of acute post-operative pain after cranial neurosurgery: A prospective cohort study. J Neurosci Rural Pract. 2023,14(4), 637–643. doi: 10.25259/JNRP_141_2023
A total of 16 characteristics were subject to statistic evaluation both in a univariate and multivariant analysis. Five of the 16 were found to have significant difference including being female, having emergency for surgery, Craniotomy and not craniectomy, operation taking over 4 hours and ASA-PS outcome measurement. The most important issues related to Female sex and younger age. The latter did not seem to be significant on table 4. The data base used in this study is likely to have the reason for the surgery such as trauma, tumor or aneurysm and what part of the calvarium is opened. Including these factors would increase the value of the study.
Thank you. Our study design was retrospective, and unfortunately, we did not collect detailed clinical information regarding the reason for surgery, type of surgery, surgical repair technique, and size of the bone flap. Previous reports have shown that postoperative CSF sepsis, CSF leak, craniotomy size, and wound infection are important risk factors for PCH. Our study did not consider these peri- and postoperative factors in its analysis, which may be a limitation of the study. Therefore, we added this as a limitation in the Discussion as follows:
Unfortunately, we did not collect detailed clinical information on perioperative and postoperative factors, including the type of surgery, surgical repair technique, size of bone flap, postoperative CSF leak, development of postoperative meningitis, and wound infection. Previous reports have shown that postoperative CSF sepsis, CSF leak, craniotomy size, and wound infection are important risk factors for PCH.
(page 12, lines 315 – page 12, lines 319)
Our study did not consider these clinical factors in its analysis, which may be a limitation.
(page 12, lines 325 – page 12, lines 326)
We also added citation as follows:
- Sezer C.; Gokten, M. ; Sezer, A.; Gezgin, I.; Binboga, A.I.; Onay, M. Role of Craniectomy Versus Craniotomy via the Retrosigmoid Approach in Decreasing the Incidence of Postoperative Headache. Int Surg 2022, 106 (1), 32–38, https://doi.org/10.9738/INTSURG-D-21-00011.1.
One of the important characters was that there was a history of headache prior to the surgery. The outcomes here are to be expected. It is known that young females prior to menopause have much higher problems with headaches especially of migraine which tends to go away with menopause. I am not certain that knowing who is likely to have the headaches may lead to different forms of treatment and plans for treatment with medication or treatment of activity etcetera. Shouldn’t there be a plan for management for all patients?
Thank you for your valuable feedback. We agree with the reviewer's comments. This study aimed to identify potential associations between age and sex and pain intensity in different surgical contexts, thereby clarifying age- and sex-related differences in postoperative pain perception. We sought to develop tailored pain management strategies based on patient demographics (age, sex) and craniotomy characteristics. For future prospective studies, we will consider including various clinical factors based on standardized protocols to derive comprehensive management plans for all patients.
The references are important with studies of a number of different reasons for surgery. From some of the references it seems as if the highest cause of headache occurs in those patients undergoing surgery of the posterior fossa. From the references it is clear that there is not an accepted protocol for post craniotomy pain. It might make the study more interesting if the authors would put a brief discussion of the next step to a protocol.
Thank you. We agree with your observation that there is currently no widely accepted protocol for management of PCH. We appreciate your suggestion to include a brief discussion of the next steps in the development of a standardized protocol. We have added this as text in the Discussion section as follows:
PCH is a condition that necessitates consideration of various clinical factors, underscoring the need for well-defined, standardized protocols in prospective studies. For instance, given existing studies that indicate a higher incidence of headaches in patients undergoing posterior fossa surgery, prospective research should incorporate standardized protocols regarding surgical site, technique, and treatment strategies.
(page 11, lines 327 – page 12, lines 332)
We have addressed all of the issues raised by the reviewers. We are grateful for the constructive comments made during the review process. We believe that our paper has been improved by implementing these suggestions.
Yours faithfully,
Jong-Hee Sohn, M.D. Ph.D.
Department of Neurology, Chuncheon Sacred Heart Hospital, Hallym University College of Medicine, 77 Sakju-ro, Chuncheon-si, Gangwon-state, 24253, Republic of Korea
Tel: +82-33-252-9970, Fax: +82-33-241-8063
E-mail: deepfoci@hallym.or.kr
Jae-June Lee M.D. Ph.D.
Departments of Anesthesiology and Pain medicine, Chuncheon Sacred Heart Hospital, Hallym University College of Medicine, 77 Sakju-ro, Chuncheon-si, Gangwon-state, 24253, Republic of Korea
Tel: +82-33-252-9970, Fax: +82-33-241-8063
E-mail: iloveu59@hallym.or.kr

Round 2
Reviewer 1 Report
Comments and Suggestions for Authors
I thank the authors for their detailed responses to my comments. The manuscript has been modified and significantly improved.
There are minor technical problems with the design of the tables, but this does not affect the final overall assessment of this article.
Author Response
July 24, 2024
Reviewer 1
Biomedicine
Dear Reviewer 1,
Please find attached a revised version of our manuscript, “Influence of age and sex on post-craniotomy headache” (biomedicine-3108820).
We thank you for your thoughtful suggestions regarding the original version of our paper; most of the suggested changes have been incorporated into the revision.
All revisions are described in detail in the order mentioned in the review, following your comments (in italics). We believe that the revisions have greatly improved the manuscript and hereby submit the revised version for publication.
Comments to author:
I thank the authors for their detailed responses to my comments. The manuscript has been modified and significantly improved.
There are minor technical problems with the design of the tables, but this does not affect the final overall assessment of this article.
We appreciate the reviewers' comments and suggestions, which helped us improve our manuscript. I have changed the design of Tables 1-5 and Appendix Tables 1-2 to conform to the journal’s guidelines.
We have addressed all of the issues raised by the reviewers. We are grateful for the constructive comments made during the review process. We believe that our paper has been improved by implementing these suggestions.
Yours faithfully,
Jong-Hee Sohn, M.D. Ph.D.
Department of Neurology, Chuncheon Sacred Heart Hospital, Hallym University College of Medicine, 77 Sakju-ro, Chuncheon-si, Gangwon-state, 24253, Republic of Korea
Tel: +82-33-252-9970, Fax: +82-33-241-8063
E-mail: deepfoci@hallym.or.kr
Jae-June Lee M.D. Ph.D.
Departments of Anesthesiology and Pain medicine, Chuncheon Sacred Heart Hospital, Hallym University College of Medicine, 77 Sakju-ro, Chuncheon-si, Gangwon-state, 24253, Republic of Korea
Tel: +82-33-252-9970, Fax: +82-33-241-8063
E-mail: iloveu59@hallym.or.kr

Reviewer 3 Report
Comments and Suggestions for Authors
The authors have dealt with my thoughts quite well. The paper now makes better sense.
Author Response
July 24, 2024
Reviewer 3
Biomedicine
Dear Reviewer 3,
Please find attached a revised version of our manuscript, “Influence of age and sex on post-craniotomy headache” (biomedicine-3108820).
We thank you for your thoughtful suggestions regarding the original version of our paper; most of the suggested changes have been incorporated into the revision.
All revisions are described in detail in the order mentioned in the review, following your comments (in italics). We believe that the revisions have greatly improved the manuscript and hereby submit the revised version for publication.
Comments to author:
The authors have dealt with my thoughts quite well. The paper now makes better sense.
We appreciate the reviewers' comments and suggestions, which helped us improve our manuscript.
We are grateful for the constructive comments made during the review process. We believe that our paper has been improved by implementing these suggestions.
Yours faithfully,
Jong-Hee Sohn, M.D. Ph.D.
Department of Neurology, Chuncheon Sacred Heart Hospital, Hallym University College of Medicine, 77 Sakju-ro, Chuncheon-si, Gangwon-state, 24253, Republic of Korea
Tel: +82-33-252-9970, Fax: +82-33-241-8063
E-mail: deepfoci@hallym.or.kr
Jae-June Lee M.D. Ph.D.
Departments of Anesthesiology and Pain medicine, Chuncheon Sacred Heart Hospital, Hallym University College of Medicine, 77 Sakju-ro, Chuncheon-si, Gangwon-state, 24253, Republic of Korea
Tel: +82-33-252-9970, Fax: +82-33-241-8063
E-mail: iloveu59@hallym.or.kr
